# Vagal Tone Differences in Empathy Level Elicited by Different Emotions and a Co-Viewer

**DOI:** 10.3390/s20113136

**Published:** 2020-06-01

**Authors:** Suhhee Yoo, Mincheol Whang

**Affiliations:** 1Department of Emotion Engineering, Sangmyung University, Seoul 03016, Korea; nuunuu05@gmail.com; 2Department of Human-Centered Artificial Intelligence, Sangmyung University, Seoul 03016, Korea

**Keywords:** empathy, vagal tone, HRV, the polyvagal theory, media, respiratory sinus arrythmia, vmHRV, co-viewer

## Abstract

Empathy can bring different benefits depending on what kind of emotions people empathize with. For example, empathy with negative emotions can raise donations to charity while empathy with positive emotions can increase participation during remote education. However, few studies have focused on the physiological differences depending on what kind of emotions people empathize with. Furthermore, co-viewer can influence the elicitation of different levels of empathy, but this has been less discussed. Therefore, this study investigated vagal response differences according to each empathy factor level elicited by different emotions and co-viewer. Fifty-nine participants were asked to watch 4 videos and to evaluate subjective valence, arousal scores, and undertake an empathy questionnaire, which included cognitive, affective and identification empathy. Half of the participants watched the videos alone and the other half watched the videos with a co-viewer. Valence and arousal scores were categorized into three levels to figure out what kind of emotions they empathized with. Empathy level (high vs. low) was determined based on the self-report scores. Two-way MANOVA revealed an interaction effect of empathy level and emotions. High affective empathy level is associated with higher vagal response regardless of what kind of emotions they empathized with. However, vagal response differences in other empathy factor level showed a different pattern depending on what kind of emotions that participant empathized with. A high cognitive empathy level showed lower vagal responses when participants felt negative or positive valence. High identification level also showed increased cognitive burden when participants empathized with negative and neutral valence. The results implied that emotions and types of empathy should be considered when measuring empathic responses using vagal tone. Two-way MANOVA revealed empathic response differences between co-viewer condition and emotion. Participants with a co-viewer felt higher vagal responses and self-reporting empathy scores only when participants empathized with arousal. This implied that the effect of a co-viewer may impact on empathic responses only when participants felt higher emotional intensity.

## 1. Introduction

Empathy, the ability to understand other’s emotion and perspective, has gathered research interest since empathy not only promotes prosocial behaviors [1] but also has social impact on various industries. For example, empathizing with negative emotions such as anger and guilt could help to raise donations to charity or increase helping behaviors [2,3,4,5]. Another study found that viewer’s behaviors, like skipping advertisements, could be reduced when people empathized with characters in the advertisements [6]. Also, the education industry anticipated that empathy could increase students’ engagement with distant education [7,8,9]. The entertainment industry consider empathy an important factor to attract audience [10]. Interestingly, effects of empathy could be different depending on what kind of emotions people empathized with. However, differences between empathy with different emotions have been less discussed.

### 1.1. Conceptualization of Empathy

It has been generally accepted that empathy consists of several factors. Among various factors, cognitive and affective (emotional) empathy have been agreed without argument. Cognitive empathy is related to perceiving or understanding others’ mental state such as emotion, feeling and thoughts [8,10,11,12,13]. Perspective taking is an example of cognitive empathy factors. Affective empathy refers to feeling others’ emotion as if it were your own emotion and sharing with others. Emotion contagion develops into affective empathy [8,14,15] and can be accompanied by physiological responses when people feel affective empathy [11]. The third factor of empathy is controversial among researchers. However, since this study focused on the empathy with multimedia content, this study considered the third component of empathy as identification. Identification refers to finding and experiencing similarities between the character’s situation and one’s experience [8,11,13,14,15,16]. Identification has been considered an important factor when measuring empathy with advertisements. Identification could make customers feel similarities with the advertisement content so that they could have a favorable response to the advertisement [8,11,14,15,16].

### 1.2. Theory of Emotion

Regarding the theoretical view of emotion, both discrete and dimensional models have generally explained emotion. The discrete model considers that there are six basic emotions that are universally experienced in all cultures. Other emotions come up with the combination of basic emotions according to the discrete emotions. On the other hand, dimensional model such as the Circumplex Model of Affect (CMA) [17,18] explains that valence and arousal dimension represent all emotions. This study classified emotion using a dimensional model to include various emotions since, in the discrete model, basic emotion is prone to explain negative emotion more without reflecting the polarity of emotion.

### 1.3. Measurement of Empathy

Empathy has been assessed through self-reporting by conceptualizing empathy. One of the reasons that self-reporting has been most frequently used is the concept of empathy is complicated, as aforementioned. Self-reporting can be a useful method to include three factors of empathy [13]. However, there is a risk it will be biased [1] and it is difficult to measure in real-time without interfering with viewers. Measuring physiological response could be an alternative.

Physiological responses have been known to reflect emotional responses [1] and can be measured in real-time. Also, this method is less biased than self-reporting. There are several physiological responses related to empathy, such as neural response. However, this study focused on vagal tone, which refers to the controlled heart responses by the vagal nerve which is the 10th cranial nerve. First, vagal tone has been studied as an index of empathy [19,20,21,22,23,24]. According to the polyvagal theory, the myelinated vagal nerve regulates peripheral autonomic activity which involves emotions, facial expression and communication [19,20,21]. The myelinated vagal fibers regulate cardiac output in response to the environment. When the environment is recognized as safe, the myelinated vagal nerve decreases cardiac output and to make the body appropriate state for the social engagement such as regulating emotion. Emotion regulation has been positively associated with empathic responses [25,26]. Second, vagal tone can be easily measured through electrocardiogram (ECG) or photoplethysmogram (PPG) than electroencephalography (EEG). Wearing EEG generally makes people feel uncomfortable and requires higher cost than ECG or PPG. This could interfere with practical uses. Facial expression has also been used to measure empathy in related studies since facial mimicry has been associated with empathy [27,28]. However, facial mimicry has been studied more in interpersonal empathy. The present study focused on empathy elicited by multimedia content. Therefore, vagal tone was chosen as an index of empathy.

### 1.4. Vagal Tone

Vagal tone has been measured through respiratory sinus arrhythmia (RSA) since vagal tone has been known to include respiratory rhythm [19,20,21,22,23,24]. RSA refers to the heart oscillation that links to respiration. Generally, the R-R interval is shortened during inspiration, while it prolonged during expiration. There are three methods to measure RSA, namely, the Porges-Bohrer method (RSA_PB), high frequency of R–R interval (HF: 0.15~0.4 Hz) and the peak-valley method. In the present study, RSA were measured using RSA_PB and HF (RSA_HF). RSA_HF is one of the most frequently used parameters when measuring RSA since HF frequency was defined by the respiration frequency [29]. RSA_PB has been known to have higher effect size compared to other methods [29]. The peak-valley method was excluded since it requires the participant to wear an additional respiratory sensor which could be a burden for participants. Another parameter of vagal tone is root mean successive difference (RMSSD). RMSSD is one of the time domain heart rate variability (HRV) and highly correlated to vagal tone [23]. Furthermore, RMSSD has short measurement time compared to other methods. It has been validated in 10-s measurements [23,30]. Therefore, it could be a good supplement considering that RSA_PB and RSA_HF requires more than 10 s of measurement. Vagal response would be more reliable when multiple vagal response parameters show a consistent pattern [31].

### 1.5. Eliciation of Empathy

Regarding elicitation of empathy, past studies have mainly focused on the level of empathy. However, different levels of empathy elicited by different emotions has been less studied, even though it could elicit different effects. Therefore, the present study tried to elicit empathy with different emotions using multimedia content. Furthermore, effect of co-viewer on eliciting empathy has been less described. Like an old saying ‘shared sorrow is sorrow halved and shared joy is joy doubled’, sharing experience such as co-viewing could impact on empathy and emotional experience during media consumption [32,33]. For example, the transfer of emotions between individuals can occur more when they co-view a movie compared to people who saw a movie alone [10]. Another study suggested higher learning may occur when students co-view multimedia content with others by increasing empathic responses [34].

To summarize, the state of the art showed the following limitations: (1) vagal response differences according to empathy level elicited by different emotions were less discussed; (2) co-viewer could influence empathy elicited by different emotions, but it has been less described. Therefore, this study investigated vagal response differences according to each empathy factor level elicited by different emotions. This study anticipated that vagal responses according to empathy level elicited by different emotions would be different unlike former studies. Former studies considered that vagal response would be higher when people felt a high level of empathy regardless of what kind of emotion they empathized with. This study also investigated empathic response differences depending on the co-viewer and what kinds of emotion participants empathized with.

## 2. Method

### 2.1. Participants

Sixty-two university students aged between 20 and 28 years without cardiovascular diseases participated the experiment voluntarily. However, 59 participants (male: 28, female: 31; age mean = 22.28; *SD* = 2.11) were included in this study due to technical failures during the experiment. They were asked to have enough sleep and caffeine intake was inhibited before the experiment. About $ 20 were paid to participants who carried out the whole experiment. Participants with a co-viewer executed the experiment with their friend.

### 2.2. Stimuli

This study tried to select stimuli which can elicit quadrants of CMA [13] to include various kinds of emotions. Five candidate videos were collected through a focus group interview before the pre-test. In the pre-test, 198 participants were asked to evaluate their emotion in terms of valence and arousal in 5-point Likert scales after they watched each stimulus. In terms of valence, they were asked to answer 1 if they felt negative and 5 if they felt positive (1: Negative~5: Positive). In case of arousal, they were asked to answer 1 if they felt relaxation and 5 if they felt arousal (1: Relaxation~5: Arousal). This process was done until they watched all five candidate videos. This study excluded the stimulus 5 after the valence and arousal mean scores of the pre-test were compared. Stimulus 5 was intended to elicit low arousal but showed similar level with high arousal. However, stimulus 5’s arousal score was similar to stimulus 2 and its valence level was higher than stimulus 4. The chosen 4-min long videos were presented to participants of the main experiment. The content of the pre-test videos was presented in Table 1.

### 2.3. Self-Report of Emotion

Participants were asked to assess their emotion in terms of the valence and arousal dimension in 5-point Likert scales after they watched each stimulus. In terms of valence, they were asked to answer 1 if they felt negative and 5 if they felt positive (1: Negative~5: Positive). In terms of arousal, they were asked to answer 1 if they felt relaxation and 5 if they felt arousal (1: Relaxation~5: Arousal). Although this study tried to elicit quadrants of CMA, it is difficult to assure that participant really felt the emotions we intended. Therefore, this study categorized participants’ assessment into three group per each emotional dimension to reflect their genuine emotions. The valence scores were categorized into three groups: 1~2 points were categorized into negative, 3 into neutral and 4~5 points were categorized into positive valence. Arousal scores were categorized in the same way (1~2: relaxation, 3: neutral, 4~5: arousal). This process was done to figure out what kind of emotions participants empathized with.

### 2.4. Self-Report of Empathy

#### 2.4.1. The Empathy Questionnaire

This study selected the questionnaire (the Consumer Empathic Response to Advertising Scale, the CERA scale) developed by a Korean researcher [13] to make sure that the effect of cultural differences could not interfere [35,36]. The questionnaire consisted of cognitive, affective and identification empathy. Each empathy has 3 to 4 items (Total items: 11). The detailed content of items was presented in Table 2. Scores of each empathy factors can be computed through averaged scores of the corresponding items.

To develop the CERA scale, [13] conducted the following steps:1.17 items from [8,11,37,38] were collected and revised through the focus group interview and the pre-test.2.225 university students were divided into two groups and they were asked to evaluate each item in 5-point scale (1: I disagree~5: I agree) after they watch advertisements. Each group saw a different advertisement. This process was done to develop the CERA scale that can be applied to various content.3.Exploratory factor analysis was conducted per each group. Only common factors and items of the results of two factor analysis were included in the CERA scale. The results revealed that empathy consisted of three factors (cognitive, affective and identification empathy) and 11 items. (All items’ factor loading exceeded 0.4, and Cronbach’s α of each factor exceeded 0.8)4.Confirmatory factor analysis was conducted to verify convergent validity of the CERA scale. In total, 225 university students were divided into two groups and they were asked to watch two different advertisements. They also were asked to answer the CERA scale in 5-point scale (1: I disagree, 5: I agree).5.The results of confirmatory factor analysis verified convergent validity and discriminant validity between each factor. Factor loadings and construct reliability of each factor exceeded 0.7. Average variance extracted (AVE) of each factor exceeded 0.5. Discriminant validity was also proved by comparing standard multiple correlation (SMC) of each pair of factors and AVE. All cases of AVE were higher than all cases of SMC.

This study verified the CERA scale by factor analysis based on the responses of 198 pre-test participants before the main experiment. Pre-test participants were asked to assess 11 items in 5-point scale in the same way as [13] described after they saw five candidate videos explained in the 2.2 stimuli section. The results revealed that the CERA scale consisted of cognitive, affective and identification empathy. KMO was over 0.9 which means that is good to use factor analysis. Bartlett’s test was significant which means correlation exists between items. Factor loadings of all items exceeded 0.6. Table 3 presented the result of verification of the CERA scale and Figure 1 showed the relationship between cognitive, affective and identification empathy.

Cog_emp_1~cog_emp_4 refers to items of cognitive empathy. Idt_emp_1~idt _emp_4 refers to items of identification and aff_emp_1~aff_emp_3 refers to items of affective empathy. Items were presented in Table 2.

#### 2.4.2. Determination of Empathy Level

In the main experiment, participants were asked to assess each item in 5-point scale (1: I disagree, 5: I agree) for empathy evaluation. Scores of each empathy factors were computed in the way described in Section 2.4.1. All types of empathy scores were categorized into two levels to assign empathy levels. Scores below the mean of each empathy scores were categorized into low level and the rest was categorized into high level. Total empathy scores were also categorized into two levels using the same method.

### 2.5. Experimental Procedure

ECG were attached to participants to collect their physiological signals during the whole experiment. In the reference session, participants asked to look at a ‘+’ mark on the center of the monitor for 4 min. After the reference session, stimuli were presented for 4 min. Participants had 3 min to answer the questionnaires after watching the stimulus. Then, the next stimulus was presented. This process was repeated until participant had watched all stimuli. The order of stimuli has been randomly presented to reduce ordering effect. The experiment procedure was approved Institutional Review Board of Sangmyung University (BE2018-35). Figure 2 and Figure 3 showed the flow chart of the experiment and experiment settings.

### 2.6. Preprocessing of Physiologcial Signal

ECG was collected using MP 100 and ECG 100c amplifier (BIOPAC System). Frequency of sampling rates was set at 500 Hz and digitized with NI-DAQ-Pad 9205 (National Instruments). Labview (National Instruments) was used to transfer the recorded signals to a computer and Python was used for data processing. R to R intervals (RRI) were calculated from raw ECG recordings and abnormal RRI were excluded from analysis. This study considered normal RRI is between 400–1333 ms. This range corresponds to 45 beat per minute (BPM) and 150 BPM. Missing RRI values were replaced by mean of former normal RRI values. Sliding window technique (window size 180 s and interval 1 s) was applied to collect signals. RMSSD data were divided by RRI to correct the heart rate. Then, the processed RMSSD data were log transformed to correct the skewness [39]. RSA_HF (0.15–0.4 Hz). RSA_HF data were normalization by RRI to reduce the effect of beta adrenergic [40,41]. The processed RSA_HF data were log transformed to correct the skewness like RSA_HF. RSA_PB was extracted according to [29]. There were no missing values of vagal responses since the missing RRI values were corrected.

### 2.7. Statistical Anaylsis

All statistical analysis was done using SPSS version 21 (IBM) and Python libraries such as Scipy, Statsmodels. Outliers were not included in the analysis. This study conducted 2 (empathy level) × 3 (emotion) multivariate analysis of variance (MANOVA) to investigate vagal differences depending on empathy level and emotion level. The Pillai’s test was conducted since Box’s M test was significant in all cases, which means that the assumption of homogeneity of covariance metrices was violated. Empathy level was measured in four factors (cognitive, affective, identification) and empathy was measured in 2 dimensions (valence and arousal). Therefore, this study conducted 2 3 MANOVA 8 times. Univariate analyses of variance (ANOVAs) and Scheffe’s post hoc test were followed to identify which variables contributed to a significant multivariate effect.

To investigate effect of co-viewers and emotion on empathic response, 2 (co-viewer condition) × 3 (emotion level) MANOVA. Empathic responses were measured in two types of variables (self-reporting and vagal tone variables) and emotion was measured in valence and arousal dimensions. Therefore, 2 × 3 MANOVA was conducted 4 times. The Wilk’s lambda test was conducted when dependent variables were empathy scores (cognitive, affective, identification and total empathy) since Box’s M test was not significant (*p* > 0.05). When dependent variables were vagal response (RSA_PB, RSA_HF, RMSSD), the Pillai’s test was conducted since Box’s M test was significant. Univariate analyses of variance (ANOVAs) and Scheffe’s post hoc test was followed to identify which variables contributed to a significant multivariate effect.

## 3. Results

### 3.1. Vagal Response Differences According to Empathy Levels Elicited by Each Emotion

#### 3.1.1. Vagal Response Differences According to Cognitive Empathy Level and Valence

Two-way MANOVA revealed vagal response differences between cognitive empathy level (Pillai’s Trace = 0.006, F (3, 14253) = 26.721, *p* < 0.001, partial *η*^2^ = 0.006). When empathy level was controlled, the effect of valence level was significant (Pillai’s Trace = 0.032, F (6, 28508) = 77.372, *p* < 0.001, partial *η*^2^ =0.016). Interaction effect of cognitive empathy level and valence level was shown to be significant (Pillai’s Trace = 0.009, F (6, 28508) = 22.021, *p* < 0.001, partial *η*^2^ = 0.006).

A separate ANOVA was conducted for each dependent variable. There was a significant difference between cognitive empathy levels on RSA_PB (F (1, 14255) = 38.739, *p* < 0.001, *η*^2^ = 0.003) and low cognitive empathy showed higher RSA_PB (M = 6.694) than high cognitive empathy (M = 6.614). There was a significant difference between cognitive empathy levels on RMSSD (F (1, 14255) = 40.69, *p* < 0.001, *η*^2^ =0.003) with low cognitive empathy showed higher RMSSD (M = −3.002) than high cognitive empathy (M = −3.047). RSA_HF between cognitive empathy level was not significant.

There was a significant difference between valence level on RSA_PB (F (2, 14255) = 66.043, *p* < 0.001, *η*^2^ = 0.009), RSA_HF (F (2, 14255) = 165.009, *p* < 0.001, *η*^2^ = 0.023) and RMSSD (F (2, 14255) = 28.108, *p* < 0.001, *η*^2^ = 0.004). Vagal responses difference pattern depending on valence level are as follows: RSA_PB (Negative > Positive > Neutral), RSA_HF (Negative > Neutral > Positive) and RMSSD (Negative >Positive > Neutral. Descriptive statistics were shown in Table 4 and Figure 4 showed the results of 2-way MANOVA.

#### 3.1.2. Vagal Response Differences According to Cognitive Empathy Level and Arousal

Two-way MANOVA revealed vagal response differences between cognitive empathy level (Pillai’s Trace = 0.002, F (3, 14253) = 8.593, *p* < 0.001, partial *η*^2^ = 0.002). When empathy level was controlled, the effect of arousal level was significant (Pillai’s Trace = 0.032, F (6, 28508) = 78.031, *p* < 0.001, partial *η*^2^ = 0.016). The interaction effect of cognitive empathy level and arousal level was shown to be significant (Pillai’s Trace = 0.004, F (6, 28508) = 10.203, *p* < 0.001, partial *η*^2^ = 0.002).

A separate ANOVA was conducted for each dependent variable. There was a significant difference between cognitive empathy levels on RSA_PB (F (1, 14255) = 16.824, *p* < 0.001, *η*^2^ = 0.001) and low cognitive empathy showed higher RSA_PB (M = 6.640) than high cognitive empathy (M = 6.518). There was a significant difference between cognitive empathy levels on RSA_HF (F (1, 14255) = 6.471, *p* < 0.011, *η*^2^ = 0) and low cognitive empathy level showed higher RSA_HF (M = 1.053) than high cognitive empathy level (M = 1.031). There was a significant difference between cognitive empathy levels on RMSSD (F (1, 14255) = 24.882, *p* < 0.001, *η*^2^ = 0.002) and low cognitive empathy showed higher RMSSD (M = −3.022) than high cognitive empathy (M = −3.06).

There was a significant difference between arousal level on RSA_PB (F (2, 14255) = 82.907, *p* < 0.001, *η*^2^ = 0.011), and RMSSD (F (2, 14255) = 73.026, *p* < 0.001, *η*^2^ = 0.010). RSA_HF differences depending on arousal level were not significant. Vagal responses difference pattern depending on arousal level are as follows: RSA_PB (Relaxation > Neutral > Arousal), and RMSSD (Relaxation >Neutral > Arousal). Descriptive statistics were shown in Table 5 and Figure 5 showed the results of 2-way MANOVA.

#### 3.1.3. Vagal Response Differences According to Affective Empathy Levels and Valence

Two-way MANOVA revealed vagal response differences between affective empathy level (Pillai’s Trace = 0.022, F (3, 14253) = 26.721, *p* < 0.001, partial *η*^2^ = 0.022). When empathy level was controlled, the effect of valence level was significant (Pillai’s Trace = 0.042, F (6, 28508) = 81.454, *p* < 0.001, partial *η*^2^ = 0.017). The interaction effect of affective empathy level and valence level was shown to be significant (Pillai’s Trace = 0.034, F (6, 28508) = 22.021, *p* < 0.001, partial *η*^2^ = 0.012). 

A separate ANOVA was conducted for each dependent variable. There was a significant difference between affective empathy levels on RSA_PB (F (1, 14255) = 265.093, *p* < 0.001, *η*^2^ = 0.018) with high affective empathy showed higher RSA_PB (M = 6.779) than low affective empathy (M = 6.561). There was a significant difference between affective empathy levels on RSA_HF (F (1, 14255) = 107.138, *p* < 0.001, *η*^2^ = 0.007) with high affective empathy showing higher RSA_HF (M = 1.097) than low affective empathy (M = 1.017). There was a significant difference between affective empathy levels on RMSSD (F (1, 14255) = 326.441, *p* < 0.001, *η*^2^ = 0.003) with high affective empathy showing higher RMSSD (M = −2.949) than high affective empathy (M = −3.077).

There was a significant difference between valence level on RSA_PB (F (2, 14261) = 93.927, *p* < 0.001, *η*^2^ = 0.013), RSA_HF (F (2, 14261) = 192.068, *p* < 0.001, *η*^2^ = 0.026) and RMSSD (F (2, 14261) = 28.757, *p* < 0.001, *η*^2^ = 0.004). Vagal responses difference pattern depending on valence level are as follows: RSA_PB (Negative > Neutral > Positive), RSA_HF (Negative > Neutral > Positive) and RMSSD (Negative >Neutral > Positive). Descriptive statistics were shown in Table 6 and Figure 6 showed the results of 2-way MANOVA.

#### 3.1.4. Vagal Response Differences According to Affective Empathy Levels and Arousal

Two-way MANOVA revealed vagal response differences between affective empathy level (Pillai’s Trace = 0.019, F (3, 14253) = 93.776, *p* < 0.001, partial *η*^2^ = 0.019). When empathy level was controlled, the effect of arousal level was significant (Pillai’s Trace = 0.037, F (6, 28508) = 90.322, *p* < 0.001, partial *η*^2^ = 0.019). The interaction effect of affective empathy level and arousal level was shown to be significant (Pillai’s Trace = 0.02, F (6, 28508) = 47.725, *p* < 0.001, partial *η*^2^ = 0.010). 

A separate ANOVA was conducted for each dependent variable. There was a significant difference between affective empathy levels on RSA_PB (F (1, 14255) = 179.585, *p* < 0.001, *η*^2^ = 0.012) with high affective empathy showed higher RSA_PB (M = 6.707) than low affective empathy (M = 6.526). There was a significant difference between affective empathy levels on RSA_HF (F (1, 14255) = 37.319, *p* < 0.001, *η*^2^ = 0.003) with high affective empathy showed higher RSA_HF (M = 1.071) than low affective empathy (M = 1.023). There was a significant difference between affective empathy levels on RMSSD (F (1, 14255) = 260.125, *p* < 0.001, *η*^2^ = 0.018) and high affective empathy showed higher RMSSD (M = −2.984) than high affective empathy (M = −3.099). 

There was a significant difference between arousal level on RSA_PB (F (2, 14261) = 128.237, *p* < 0.001, *η*^2^ = 0.018), RSA_HF (F (2, 14261) = 5.934, *p* < 0.001, *η*^2^ = 0.001) and RMSSD (F (2, 14261) = 115.41, *p* < 0.001, *η*^2^ = 0). Vagal responses difference pattern depending on arousal level are as follows: RSA_PB (Relaxation > Neutral > Arousal), RSA_HF (Relaxation > Arousal > Neutral) and RMSSD (Relaxation >Neutral > Arousal). Descriptive statistics were shown in Table 7 and Figure 7 showed the results of 2-way MANOVA.

#### 3.1.5. Vagal Response Differences According to Identification Levels and Valence

Two-way MANOVA revealed vagal response differences between identification empathy level (Pillai’s Trace = 0.011, F (3, 14253) = 52.326, *p* < 0.001, partial *η*^2^ = 0.011). When empathy level was controlled, the effect of valence level was significant (Pillai’s Trace = 0.034, F (6, 28508) = 82.75, *p* < 0.001, partial *η*^2^ = 0.017). Interaction effect of identification empathy level and valence level was shown to be significant (Pillai’s Trace = 0.014, F (6, 28508) = 3.056, *p* < 0.001, partial *η*^2^ = 0.007). 

A separate ANOVA was conducted for each dependent variable. There was a significant difference between identification empathy levels on RSA_PB (F (1, 14255) = 63.034, *p* < 0.001, *η*^2^ = 0.004) and low identification empathy showed higher RSA_PB (M = 6.707) than high identification empathy (M = 6.597). There was a significant difference between identification empathy levels on RMSSD (F (1, 14255) = 41.140, *p* < 0.001, *η*^2^ = 0.003) with low identification empathy showed higher RMSSD (M = −3.005) than high identification empathy (M = −3.05). RSA_HF differences according to identification level was not significant.

There was a significant difference between valence level on RSA_PB (F (2, 14261) = 75.806, *p* < 0.001, *η*^2^ = 0.011), RSA_HF (F (2, 14261) = 183.28, *p* < 0.001, *η*^2^ = 0.025) and RMSSD (F (2, 14261) = 30.817, *p* < 0.001, *η*^2^ = 0.004). Vagal responses difference pattern depending on valence level are as follows: RSA_PB (Negative > Neutral > Positive), RSA_HF (Negative > Neutral > Positive) and RMSSD (Negative >Neutral > Positive). Descriptive statistics were shown in Table 8 and Figure 8 showed the results of 2-way MANOVA.

#### 3.1.6. Vagal Response Differences According to Identification Levels and Arousal

Two-way MANOVA revealed vagal response differences between identification empathy level (Pillai’s Trace = 0.004, F (3, 14253) = 17.894, *p* < 0.001, partial *η*^2^ = 0.004). When empathy level was controlled, the effect of arousal level was significant (Pillai’s Trace = 0.032, F (6, 28508) = 78.443, *p* < 0.001, partial *η*^2^ = 0.016). Interaction effect of identification empathy level and arousal level was shown to be significant (Pillai’s Trace = 0.018, F (6, 28508) = 44.355, *p* < 0.001, partial *η*^2^ = 0.009).

A separate ANOVA was conducted for each dependent variable. There was a significant difference between identification levels on RSA_HF (F (1, 14255) = 16.136, *p* < 0.001, *η*^2^ = 0.001) with high identification showed higher RSA_PB (M = 1.043) than low identification (M = 1.009). RSA_PB and RMSSD differences according to identification level were not significant.

There was a significant difference between arousal level on RSA_PB (F (2, 14261) = 59.266, *p* < 0.001, *η*^2^ = 0.008), RSA_HF (F (2, 14261) = 107.255, *p* < 0.001, *η*^2^ = 0.015) and RMSSD (F (2, 14261) = 37.604, *p* < 0.001, *η*^2^ = 0.005). Vagal responses difference pattern depending on arousal level are as follows: RSA_PB (Relaxation > Neutral > Arousal), RSA_HF (Relaxation > Neutral > Arousal) and RMSSD (Relaxation >Neutral > Arousal). Descriptive statistics were shown in Table 9 and Figure 9 showed the results of 2-way MANOVA.

#### 3.1.7. Vagal Response Differences According to Total Empathy Levels and Valence

Two-way MANOVA revealed vagal response differences between total empathy level (Pillai’s Trace = 0.005, F (3, 14253) = 21.814, *p* < 0.001, partial *η*^2^ = 0.005). When empathy level was controlled, the effect of valence level was significant (Pillai’s Trace = 0.036, F (6, 28508) = 87.11, *p* < 0.001, partial *η*^2^ = 0.018). Interaction effect of total empathy level and valence level was shown to be significant (Pillai’s Trace = 0.009, F (6, 28508) = 21.138, *p* < 0.001, partial *η*^2^ = 0.004). 

A separate ANOVA was conducted for each dependent variable. There was a significant difference between total empathy levels on RSA_HF (F (1, 14255) = 24.727, *p* < 0.001, *η*^2^ = 0.002) with high total empathy showed higher RSA_HF (M = 1.08) than low total empathy (M = 1.041). There was a significant difference between total empathy levels on RMSSD (F (1, 14255) = 16.333, *p* < 0.001, *η*^2^ = 0.001) with high total empathy showed higher RMSSD (M = −3.008) than high total empathy (M = −3.037). RSA_PB differences according to total empathy level were not significant.

There was a significant difference between valence level on RSA_PB (F (2, 14261) = 98.819, *p* < 0.001, *η*^2^ = 0.014), RSA_HF (F (2, 14261) = 203.727, *p* < 0.001, *η*^2^ = 0.028) and RMSSD (F (2, 14261) = 39.772, *p* < 0.001, *η*^2^ = 0.006). Vagal responses difference pattern depending on valence level are as follows: RSA_PB (Negative > Neutral > Positive), RSA_HF (Negative > Neutral > Positive) and RMSSD (Negative >Neutral > Positive). Descriptive statistics were shown in Table 10 and Figure 10 shows the results of 2-way MANOVA.

#### 3.1.8. Vagal Response Differences According to Total Empathy Levels and Arousal

Two-way MANOVA revealed vagal response differences between total empathy level (Pillai’s Trace = 0.004, F (3, 14253) = 20.377, *p* < 0.001, partial *η*^2^ = 0.004). When empathy level was controlled, the effect of arousal level was significant (Pillai’s Trace = 0.028, F (6, 28508) = 78.443, *p* < 0.001, partial *η*^2^ = 0.014). Interaction effect of total empathy level and arousal level was shown to be significant (Pillai’s Trace = 0.013, F (6, 28508) = 31.772, *p* < 0.001, partial *η*^2^ = 0.007). 

A separate ANOVA was conducted for each dependent variable. There was a significant difference between total levels on RMSSD (F (1, 14255) = 12.757, *p* < 0.001, *η*^2^ = 0.001) and high total empathy showed higher RMSSD (M = −3.026) than low total empathy (M = −3.053). RSA_PB and RSA_HF differences according to total level was not significant.

There was a significant difference between arousal level on RSA_PB (F (2, 14261) = 88.874, *p* < 0.001, *η*^2^ = 0.012), RSA_HF (F (2, 14261) = 6.258, *p* < 0.001, *η*^2^ = 0.001) and RMSSD (F (2, 14261) = 81.847, *p* < 0.001, *η*^2^ = 0.011). Vagal responses difference pattern depending on arousal level are as follows: RSA_PB (Relaxation > Neutral > Arousal), RSA_HF (Relaxation > Arousal > Neutral) and RMSSD (Relaxation >Neutral > Arousal). Descriptive statistics were shown in Table 11 and Figure 11 showed the results of 2-way MANOVA.

### 3.2. Effect of Co-Viewers on Empathy

#### 3.2.1. Self Report Differences According to Co-Viewer Condition and Valence

Two-way MANOVA revealed empathy score differences between participants with co-viewer and without co-viewer (Wilk’s λ = 0.097, F (4, 227) = 5.941, *p* < 0.001, partial *η*^2^ = 0.095). When co-viewer condition was controlled, the effect of valence level was significant (Wilk’s λ = 0.036, F (8, 456) = 4.118, *p* < 0.001, partial *η*^2^ = 0.067). Interaction effect of total empathy level and valence level was shown to be significant (Wilk’s λ = 0.059, F (8, 456) = 1.742, *p* < 0.001, partial *η*^2^ = 0.03).

A separate ANOVA was conducted for each dependent variable, with each ANOVA evaluated at alpha level of 0.008. There was a significant difference between participants with co-viewer and without co-viewer on affective empathy scores (F (1, 230) = 24.727, *p* < 0.001, *η*^2^ =0.002) and participants with co-viewer showed higher affective empathy scores (M = 3.019) than participants without co-viewer (M = 2.318). There was a significant difference between participants with co-viewer and without co-viewer on identification (F (1, 230) = 16.333, *p* < 0.001, *η*^2^ = 0.001) with participants with co-viewer showed higher identification score (M = 3.489) than participants without co-viewer (M = 3.044). There was a significant difference between participants with co-viewer and without co-viewer on total empathy scores (F (1, 230) = 16.333, *p* < 0.001, *η*^2^ = 0.001) and participants with co-viewer showed higher total empathy score (M = 3.683) than participants without co-viewer (M = 3.261). Cognitive empathy score differences according to co-viewer condition was not significant.

There was a significant difference between valence level on affective empathy (F (2, 230) = 9.294, *p* < 0.001, *η*^2^ = 0.075), identification (F (2, 230) = 7.243, *p* < 0.001, *η*^2^ = 0.059) and total empathy (F (2, 230) = 10.067, *p* < 0.001, *η*^2^ = 0.080). Cognitive empathy score differences according to valence level was not significant. Each empathy score differences depending on valence level are as follows: affective empathy (Positive > Negative > Neutral), identification (Positive > Neutral > Negative) and total empathy (Positive > Negative > Neutral). Descriptive statistics were shown in Table 12 and Figure 12 showed the results of 2-way MANOVA.

#### 3.2.2. Self Report Differences According to Co-Viewer Condition and Arousal

Two-way MANOVA revealed empathy score differences between participants with co-viewer and without co-viewer (Wilk’s λ= 0.128, F (4, 227) = 5.941, *p* < 0.001, partial *η*^2^ = 0.128). When co-viewer condition was controlled, the effect of arousal level was significant (Wilk’s λ = 0.036, F (8, 456) = 3.476, *p* < 0.001, partial *η*^2^ = 0.057). The interaction effect of total empathy level and arousal level was shown to be significant (Wilk’s λ = 0.059, F (8, 456) = 0.572, *p* < 0.001, partial *η*^2^ = 0.010).

A separate ANOVA was conducted for each dependent variable, with each ANOVA evaluated at alpha level of 0.008. There was a significant difference between participants with co-viewer and without co-viewer on affective empathy scores (F (1, 230) = 29.426, *p* < 0.001, *η*^2^ = 0.113) with participants with co-viewer showed higher affective empathy scores (M = 3.297) than participants without co-viewer (M = 2.423). There was a significant difference between participants with co-viewer and without co-viewer on identification (F (1, 230) = 12.904, *p* < 0.001, *η*^2^ = 0.001) and participants with co-viewer showed higher identification score (M = 3.684) than participants without co-viewer (M = 3.113). There was a significant difference between participants with co-viewer and without co-viewer on total empathy scores (F (1, 230) = 10.083, *p* < 0.001, *η*^2^ = 0.001) and participants with co-viewer showed higher total empathy score (M = 3.862) than participants without co-viewer (M = 3.328). Cognitive empathy score differences according to co-viewer condition was not significant.

There was a significant difference between arousal level on cognitive empathy (F (2, 230) = 7.65, *p* < 0.001, *η*^2^ = 0.062), affective empathy (F (2, 230) = 8.105, *p* < 0.001, *η*^2^ = 0.066) and total empathy (F (2, 230) = 10.083, *p* < 0.001, *η*^2^ = 0.081). Identification score differences according to arousal level were not significant. Each empathy score differences depending on arousal level are as follows: cognitive empathy (Arousal > Relaxation > Neutral), affective empathy (Arousal > Relaxation > Neutral) and total empathy (Arousal > Relaxation > Neutral). Descriptive statistics were shown in Table 13 and Figure 13 showed the results of 2-way MANOVA.

#### 3.2.3. Vagal Response Differences According to Co-Viewer and Valence

Two-way MANOVA revealed vagal response differences between participants with co-viewer and without co-viewer (Pillai’s Trace = 0.10, F (3, 14253) = 47.425, *p* < 0.001, partial *η*^2^ = 0.010). When co-viewer condition was controlled, the effect of valence level was significant (Pillai’s Trace = 0.033, F (6, 28508) = 78.495, *p* < 0.001, partial *η*^2^ = 0.016). Interaction effect of co-viewer and valence level was shown to be significant (Pillai’s Trace = 0.008, F (6, 28508) = 18.381, *p* < 0.001, partial *η*^2^ = 0.004).

A separate ANOVA was conducted for each dependent variable. There was a significant difference between participants with co-viewer and without co-viewer on RSA_HF (F (1, 14255) = 59.282, *p* < 0.001, *η*^2^ = 0.004) with participants with co-viewer showed higher RSA_HF (M = 1.089) than participants without co-viewer (M = 1.031). RSA_PB and RMSSD differences according to co-viewer condition were not significant. 

There was a significant difference between valence level on RSA_PB (F (2, 14261) = 87.209, *p* < 0.001, *η*^2^ =0.012), RSA_HF (F (2, 14261) = 162.748, *p* < 0.001, *η*^2^ = 0.022) and RMSSD (F (2, 14261) = 34.784, *p* < 0.001, *η*^2^ = 0.005). Vagal responses difference pattern depending on valence level are as follows: RSA_PB (Negative > Neutral > Positive), RSA_HF (Negative > Neutral > Positive) and RMSSD (Negative >Positive > Neutral). Descriptive statistics were shown in Table 14 and Figure 14 showed the results of 2-way MANOVA.

#### 3.2.4. Vagal Response Differences According to Co-Viewer and Arousal

Two-way MANOVA revealed vagal response differences between participants with co-viewer and without co-viewer (Pillai’s Trace = 0.008, F (3, 14253) = 38.098, *p* < 0.001, partial *η*^2^ = 0.008). When co-viewer condition was controlled, the effect of arousal level was significant (Pillai’s Trace = 0.036, F (6, 28508) = 87.859, *p* < 0.001, partial *η*^2^ = 0.018). Interaction effect of co-viewer and arousal level was shown to be significant (Pillai’s Trace = 0.044, F (6, 28508) = 18.381, *p* < 0.001, partial *η*^2^ = 0.022).

A separate ANOVA was conducted for each dependent variable. There was a significant difference between participants with co-viewer and without co-viewer on RSA_HF (F (1, 14255) = 23.194, *p* < 0.001, *η*^2^ = 0.002) with participants with co-viewer showed higher RSA_HF (M = 1.067) than participants without co-viewer (M = 1.028). Co-viewer effect on RMSSD was significant (F (1, 14255) = 13.35, *p* < 0.001, *η*^2^ = 0.001). Participants with co-viewer showed higher RMSSD (M = −3.051) than participants without co-viewer (M = 1.031). RSA_PB differences according to co-viewer condition was not significant.

There was a significant difference between arousal level on RSA_PB (F (2, 14261) = 25.411, *p* < 0.001, *η*^2^ = 0.004), RSA_HF (F (2, 14261) = 43.934, *p* < 0.001, *η*^2^ = 0.006) and RMSSD (F (2, 14261) = 57.56, *p* < 0.001, *η*^2^ = 0.008). Vagal responses difference pattern depending on arousal level are as follows: RSA_PB (Relaxation > Neutral > Arousal), RSA_HF (Relaxation > Arousal > Neutral) and RMSSD (Relaxation > Neutral > Arousal). Descriptive statistics were shown in Table 15 and Figure 15 showed the results of 2-way MANOVA.

## 4. Discussion

### 4.1. Difference of Vagal Responses According to Each Empathy Level Elicited by Different Emotions

This study investigated vagal response differences depending on each empathy levels when participants empathized with different emotions. Two-way MANOVA revealed that all empathy factors showed an interaction effect with valence and arousal level. This result suggested what kinds of emotion participants empathized with should be considered when measuring empathic response through vagal tone.

Among empathy factors, only affective empathy level difference showed consistent vagal responses pattern regardless of what kind of emotions participants empathized with. High affective empathy level showed association with higher vagal responses. There was no conflict between vagal response parameters when participants empathized with different valence levels. When participants empathized with different arousal levels, high affective empathy level showed higher RSA_PB and RMSSD. Two vagal response parameters out of three showed consistent results. Therefore, it is likely that higher affective empathy level showed higher vagal responses regardless of what kind of valence and arousal levels participants empathized with. The current results are concurrent with former studies, which reported that higher vagal tone is related to higher empathy level [21,23,42]. However, this study found that vagal tone pattern is more evident when participants empathized with negative valence and low level of arousal. According to individual results of ANOVAs after two-way MANOVA, vagal differences were larger when participants empathized with negative and relaxation. Although the vagal response patterns were not different according to what kinds of emotion participants empathized with, the results of the present study showed vagal tone could detect not only the level of affective empathy but also kinds of emotion individuals empathized with.

On the other hand, vagal response differences according to cognitive, identification and total empathy level showed different patterns according to what kind of emotions participants empathized with. Also, statistically insignificant vagal responses increased compared to affective empathy level differences.

Low cognitive empathy level showed association with higher vagal tone when participants empathized with positive and negative valence since the results showed higher RSA_PB, and RMSSD. On the other hand, high cognitive empathy level showed higher vagal tone (RSA_PB and RSA_HF) when participants empathized with neutral valence. Lower cognitive empathy level showed higher vagal tone (RSA_PB, RMSSD) when participants empathized with relaxation. Previous studies have reported that lower vagal tone (RMSSD, RSA_HF) reflects more cognitive load [21,23,41]. Therefore, when participants felt negative or positive valence, it is likely to increase cognitive load than neutral valence. Also, a former study which investigated the association between empathy and arousal suggested that high arousal can increase personal distress [43]. Considering that cognitive empathy reflects more cognitive process than affective empathy, it is concurrence with the present result that high cognitive empathy level showed lower vagal responses when the participant felt negative or positive valence. Low cognitive empathy showed an association with higher vagal tone when participants empathized with relaxation compared to arousal. However, the vagal tone differences are small according to cognitive empathy level.

High identification level showed higher vagal tone when they empathized with positive valence. On the other hand, when participants empathized with neutral and negative valence, high identification level showed lower vagal tone. When participants empathized with arousal, high identification showed higher vagal tone while low level of identification showed higher vagal response when participants empathized with neutral level of arousal. Interestingly, there was no conflict between vagal parameters. Identification has cognitive and affective aspects simultaneously, as it is recalling one’s own experience and emotion [12,44]. Therefore, high identification level may also increase cognitive burden when a participant empathized with negative and neutral valence. However, cognitive burden could be different when participants empathized with positive valence unlike cognitive empathy. When participants empathized with positive valence, it seemed that cognitive aspects had less influence on vagal responses.

High total empathy level showed higher vagal responses (RSA_HF, RMSSD) when participants empathized with positive and negative valence. Only RMSSD showed vagal differences according to total empathy level when participants empathized with different levels of arousal. This implied that total empathy level may not be associated with vagal tone when participants felt different levels of arousal. Overall, total empathy level showed less significant vagal responses parameters compared to other empathy factors. This finding indicated that types of empathy level that can be detected by vagal tone is limited.

### 4.2. Effect of Co-Viewers on Empathy and Vagal Response Elicited by Various Emotions

This study investigated self-report empathy scores and vagal response differences according to co-viewer condition and what kind of emotion participants empathized. Two-way MANOVA were conducted twice and the results were compared.

In the case of self-report empathy scores, participants with co-viewer showed higher affective, identification and total empathy scores regardless of what kind of emotions participants empathized with. Cognitive empathy was not significant according to co-viewer condition. The result indicated that participants are likely to evaluate their empathy higher when they watch media content with a co-viewer. On the other hand, vagal responses showed different results. RSA_PB did not show significant differences according to co-viewer conditions in all cases. RMSSD also showed insignificant according to co-viewer condition when participants felt a different level of valence. Only RSA_HF showed significant differences according to co-viewer condition. The results indicate that co-viewer condition may not affect vagal responses, especially when the participant empathized with a different valence level. When participants felt a different level of arousal, vagal response (RSA_HF and RMSSD) were significant according to co-viewer condition and arousal levels. When we compared the results of self-reporting empathy scores and vagal responses, participants with co-viewer showed higher empathic response only when participants empathized with arousal. Other cases showed a conflicting result between self-reporting empathy scores and vagal responses. The result suggested that co-viewing may impact on empathic responses only when the intensity of emotion is high considering that subjective arousal can be interpreted as higher intensity of emotion [44,45]. When participants felt lower emotional intensity, a co-viewer did not influence empathic responses.

### 4.3. Limitations

The present study measured three vagal tone parameters for measurement stability of vagal tone [31]. Overall, vagal tone parameters were consistent or at least two out of three parameters showed consistent results. If conflicts between vagal tone parameters occurred, vagal responses were regarded as being weaker compared to cases that vagal tone parameters were consistent. However, there is a possibility that certain vagal tone would be more appropriate than others in certain cases. In future studies, this issue should be considered.

## 5. Conclusions

This study aimed to investigate vagal tone differences according to empathy levels and what kind of emotions participants empathized with. Two-way MANOVA (empathy levels × emotion) revealed that vagal tone was affected by empathy levels and what kinds of emotion participants empathized with. Affective empathy level showed consistent vagal responses regardless of what kind of emotions participants empathized with. High affective empathy level showed higher vagal response. However, empathy factors such as cognitive, identification and total empathy showed different vagal response pattern depending on what kind of emotions that participant empathized with. The results implied that emotions and types of empathy should be considered when measuring empathic responses using vagal tone. This study also aimed to investigate the effect of co-viewer and emotion since a co-viewer could impact on empathy. Two-way MANOVA (co-viewer condition × emotion) revealed empathic response differences between co-viewer condition and emotion. Participants with a co-viewer felt higher vagal responses and self-reporting empathy scores only when participants empathized with arousal. This implied that the effect of a co-viewer may impact on empathic responses only when participants felt higher emotional intensity.

## Figures and Tables

**Figure 1 sensors-20-03136-f001:**
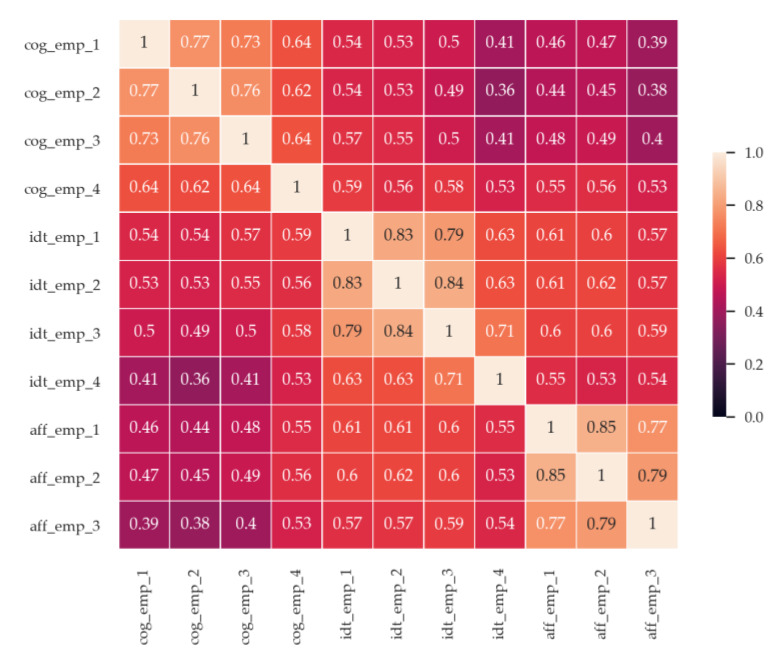
Correlation matrix of the CERA scale.

**Figure 2 sensors-20-03136-f002:**
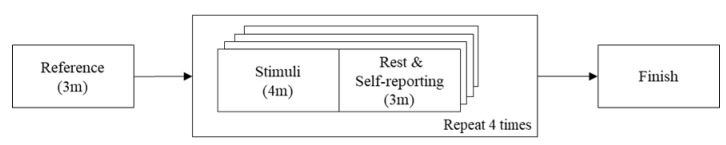
Flow chart of the experiment.

**Figure 3 sensors-20-03136-f003:**
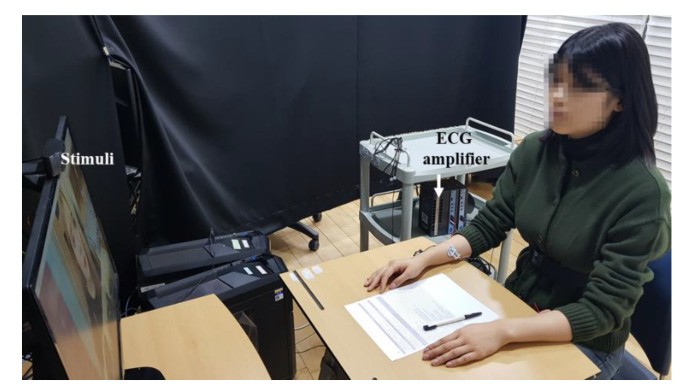
Experimental settings for participants with no co-viewer. Participants with co-viewer sit next to each other and shared the same display.

**Figure 4 sensors-20-03136-f004:**
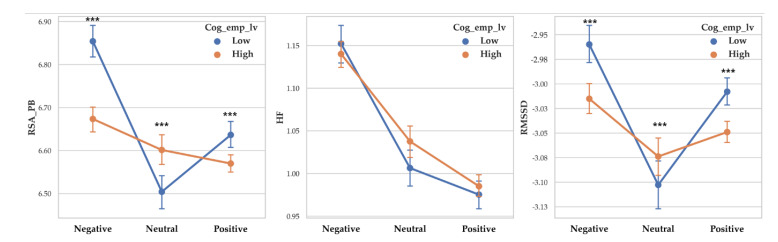
Point graph of vagal responses depending on cognitive empathy level and valence. Cog_emp_lv refers to cognitive empathy level. Point means mean of vagal responses and error bars represent confidence interval. Asterisk means significant vagal responses difference according to cognitive empathy level. (***: *p* < 0.001).

**Figure 5 sensors-20-03136-f005:**
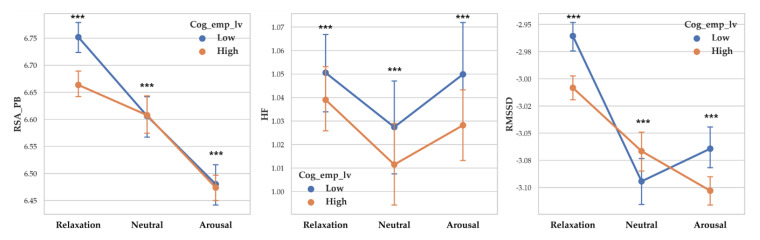
Point graph of vagal response depending on cognitive empathy level and arousal. Cog_emp_lv refers to cognitive empathy level. Point means mean of vagal responses and error bars represent confidence interval. Asterisk means significant vagal responses difference according to cognitive empathy level. (***: *p* < 0.001).

**Figure 6 sensors-20-03136-f006:**
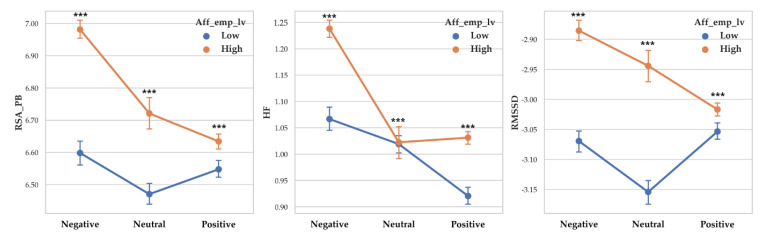
Point graph of vagal response depending on affective empathy level when participant empathized with valence levels. Aff_emp_lv refers to affective empathy level. Point means mean of vagal responses and error bars represent confidence interval. Asterisk means significant vagal responses difference according to affective empathy level. (***: *p* < 0.001).

**Figure 7 sensors-20-03136-f007:**
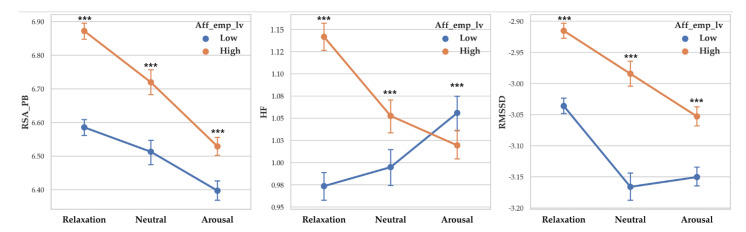
Point graph of vagal response depending on affective empathy level when participant empathized with arousal levels. Aff_emp_lv refers to affective empathy level. Point means mean of vagal responses and error bars represent confidence interval. Asterisk means significant vagal responses difference according to affective empathy level. (***: *p* < 0.001).

**Figure 8 sensors-20-03136-f008:**
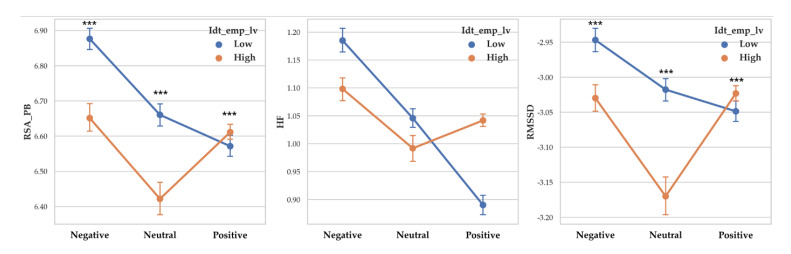
Point graph of vagal responses depending on identification empathy level when participant empathized with each valence level. Idt_emp_lv refers to identification empathy level. Point means mean of vagal responses. Asterisk means significant vagal responses difference according to identification empathy level. (***: *p* < 0.001).

**Figure 9 sensors-20-03136-f009:**
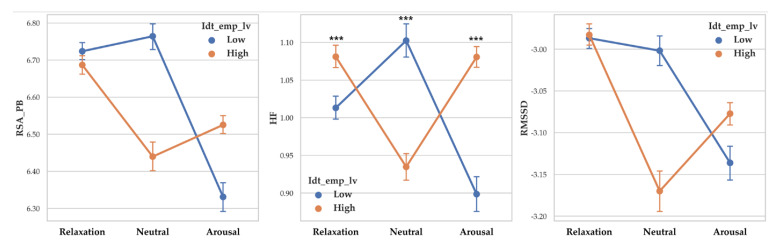
Point graph of vagal response depending on identification empathy level when participant empathized with arousal levels. Idt_emp_lv refers to identification level. Point means mean of vagal responses. Asterisk means significant vagal responses difference according to identification empathy level. (***: *p* < 0.001).

**Figure 10 sensors-20-03136-f010:**
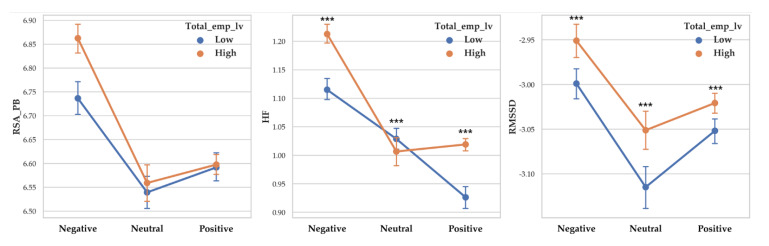
Point graph of vagal response depending on total empathy level when participant empathized with valence. Total_emp_lv refers to total empathy level. Point means mean of vagal responses. Asterisk means significant vagal responses difference according to total empathy level. (***: *p* < 0.001).

**Figure 11 sensors-20-03136-f011:**
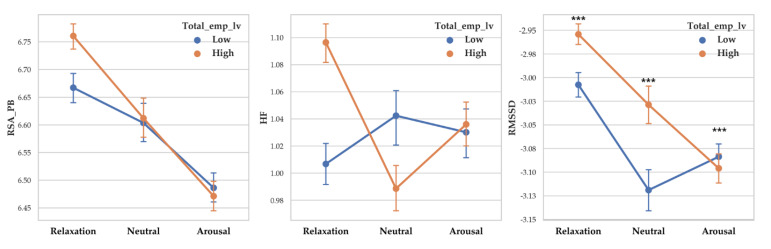
Point graph of vagal response depending on identification empathy level when participant empathized with arousal levels. Total_emp_lv refers to total empathy level. Point means mean of vagal responses. Asterisk means significant vagal responses difference according to cognitive empathy level. (***: *p* < 0.001).

**Figure 12 sensors-20-03136-f012:**
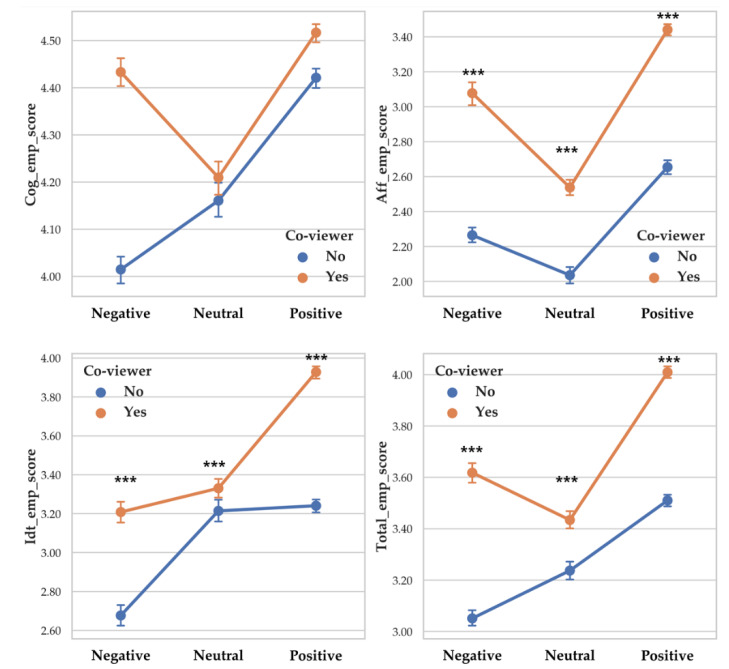
Point graph of self-report empathy depending on co-viewer condition and valence levels. Point means mean of vagal responses. ‘Co-viewer no’ means participants without co-viewer. ‘Co-viewer yes ‘means participants with co-viewer. Asterisk means significant empathy score difference according to co-viewer condition. (***: *p* < 0.001).

**Figure 13 sensors-20-03136-f013:**
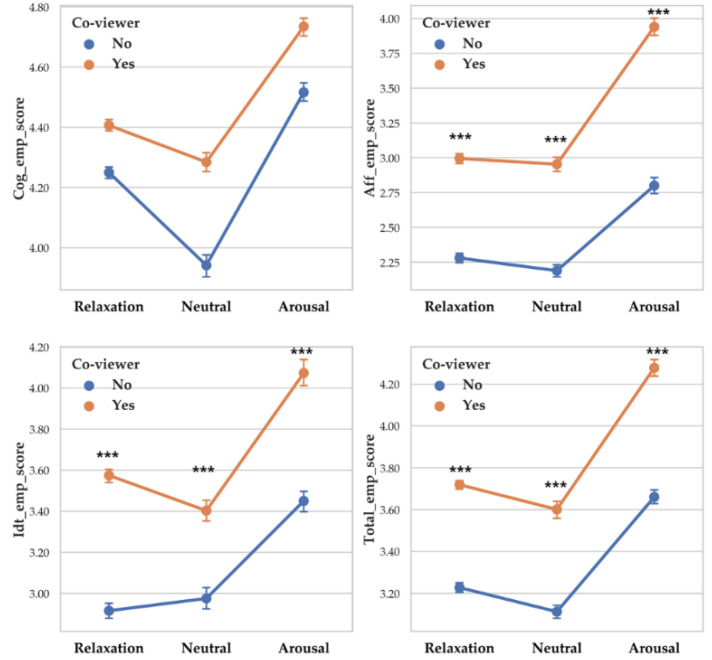
Point graph of self-report empathy depending on co-viewer condition and arousal level. ‘Co-viewer no’ means participants without co-viewer. ‘Co-viewer yes’ means participants with co-viewers. Asterisk means significant empathy score difference according to co-viewer condition. (***: *p* < 0.001).

**Figure 14 sensors-20-03136-f014:**
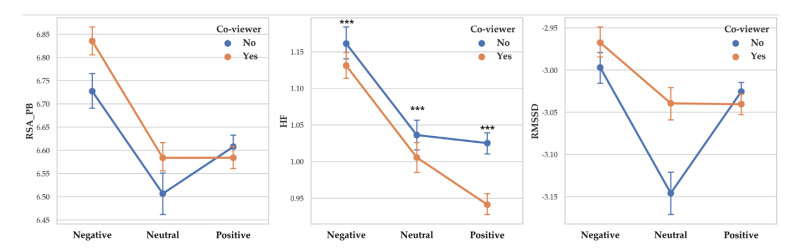
Point graph of vagal response depending on co-viewer condition and arousal level. Point means mean of vagal responses. Asterisk means significant vagal responses difference according to co-viewer condition. (***: *p* < 0.001).

**Figure 15 sensors-20-03136-f015:**
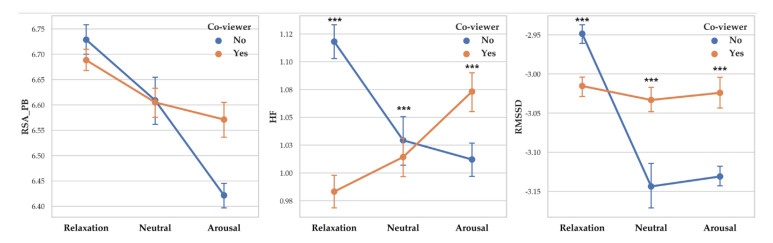
Point graph of vagal response depending on co-viewers and arousal levels. Point means mean of vagal responses. Asterisk means significant vagal responses difference according to co-viewer condition. (***: *p* < 0.001).

**Table 1 sensors-20-03136-t001:** Content of pre-test stimuli. Mean scores of each dimension also presented. Stimulus 5 was not included in the main experiment.

Stimuli	Content	Valence (M ± SD) ^1^	Arousal (M ± SD)	Targeted Emotion
1	Highlight of broadcast of soccer game when the national team won a come- from-behind victory.	4.12 ± 0.74	4.03 ± 0.80	Positive valence & High arousal
2	A scene of a girl who ate blue colored food. She intentionally put blue coloring into the normal food and described the feeling before and after she put blue coloring	2.04 ± 0.99	3.35 ± 0.89	Negative valence & High arousal
3	The moment that deaf babies hear sound for the first time.	4.10 ± 0.88	2.58 ± 1.10	Positive valence & Low arousal
4	The monologue of a man who just broke up with his girlfriend.	2.67 ± 0.87	2.85 ± 0.85	Negative valence & Low arousal
5	A scene in which characters suffer from drifting at sea for a long time.	2.71 ± 0.80	3.34 ± 0.86	Negative valence & Low arousal

^1^ M = mean, SD = standard deviation.

**Table 2 sensors-20-03136-t002:** Items of the Consumer Empathic Response to Advertising Scale (CERA) (empathy questionnaire).

Factors of Empathy	Items
Cognitive empathy	1. I understood the characters’ desire 2. I understood how character is feeling 3. I understood the situation of the content 4. I understood the character’s motive behind their behavior
Affective empathy	1. I felt the events in the content were really happening to me 2. I felt as if I was in the middle of the situation 3. I felt as if I was one of the characters
Identification empathy	1. I experienced many of the same feelings that characters portrayed 2. I experienced desires of the characters which were similar with my current desire or desire that I could feel in the future 3. I experienced the events in the content were similar with my experience or experience that could happen to me 4. I felt as if the events in the content could happen to me
Total empathy	All items were included

Total empathy score can be calculated by averaging all of the items’ scores.

**Table 3 sensors-20-03136-t003:** Verification of the CERA scale using responses of 198 pre-test participants.

Empathy Factors and Items ^1^	Factor Analysis					Reliability
	Factor Loading	Comm. ^2^	Eigen Value	% of Var. ^3^	Cum.Var % ^4^	Cronbach’s α
Cognitive empathy			6.802	28.467	28.467	0.898
1	0.850	0.815				
2	0.878	0.844				
3	0.836	0.807				
4	0.635	0.665				
Identification			1.329	28.351	56.818	0.916
1	0.770	0.819				
2	0.799	0.844				
3	0.839	0.877				
4	0.753	0.698				
Affective empathy			0.845	24.790	81.608	0.925
1	0.833	0.874				
2	0.845	0.889				
3	0.839	0.845				
KMO	0.921					
Bartlett’s test of sphericity	Approximate Chi-square			9415.282		
	df (*p*) ^5^			55 (0.000)		

^1^ The content of items was presented in Table 2. ^2^ Comm. refers to communality. ^3^ % of Var. refers to percentage of variance explained. ^4^ Cum.Var % refers to cumulative variance explained. ^5^ df refers to degree of freedom and (*p*) refers to *p*-value.

**Table 4 sensors-20-03136-t004:** Descriptive statistics of vagal response depending on cognitive empathy level and valence level. M = Mean, SD = Standard deviation, 95% CI = 95% of confidence interval.

Vagal Response	Cognitive Empathy Level	Valence Level	M	95% CI	
				Lower	Upper
RSA_PB	High	Negative	6.67	6.64	6.71
		Neutral	6.60	6.56	6.64
		Positive	6.57	6.54	6.59
	Low	Negative	6.85	6.82	6.88
		Neutral	6.60	6.56	6.63
		Positive	6.64	6.61	6.67
RSA_HF	High	Negative	1.14	1.12	1.16
		Neutral	1.04	1.02	1.06
		Positive	1.00	0.99	1.01
	Low	Negative	1.15	1.13	1.17
		Neutral	1.04	1.02	1.06
		Positive	0.98	0.96	0.99
RMSSD	High	Negative	−3.02	−3.03	−3.00
		Neutral	−3.07	−3.09	−3.05
		Positive	−3.05	−3.06	−3.04
	Low	Negative	−2.96	−2.98	−2.94
		Neutral	−3.04	−3.06	−3.02
		Positive	−3.01	−3.02	−2.99

**Table 5 sensors-20-03136-t005:** Descriptive statistics of vagal response depending on cognitive empathy level and arousal level. M = Mean, SD = Standard deviation, 95% CI = 95% of confidence interval.

Vagal Response	Cognitive Empathy Level	Arousal Level	M	95% CI	
				Lower	Upper
RSA_PB	High	Arousal	6.47	6.44	6.51
		Neutral	6.48	6.43	6.53
		Relaxation	6.61	6.57	6.64
	Low	Arousal	6.69	6.66	6.72
		Neutral	6.66	6.64	6.68
		Relaxation	6.75	6.73	6.78
RSA_HF	High	Arousal	1.03	1.01	1.05
		Neutral	1.05	1.02	1.08
		Relaxation	1.01	0.99	1.03
	Low	Arousal	1.06	1.04	1.07
		Neutral	1.05	1.04	1.07
		Relaxation	1.05	1.04	1.06
RMSSD	High	Arousal	−3.10	−3.12	−3.09
		Neutral	−3.06	−3.09	−3.04
		Relaxation	−3.07	−3.09	−3.05
	Low	Arousal	−3.04	−3.06	−3.03
		Neutral	−3.01	−3.02	−3.00
		Relaxation	−2.96	−2.97	−2.95

**Table 6 sensors-20-03136-t006:** Descriptive statistics of vagal response depending on affective empathy level and valence. M = Mean, SD = Standard deviation. 95% CI = 95% of confidence interval.

Vagal Response	Affective Empathy Level	Valence Level	M	95% CI	
				Lower	Upper
RSA_PB	High	Negative	6.98	6.95	7.02
		Neutral	6.72	6.68	6.76
		Positive	6.63	6.61	6.66
	Low	Negative	6.60	6.57	6.63
		Neutral	6.54	6.51	6.57
		Positive	6.54	6.52	6.57
RSA_HF	High	Negative	1.24	1.22	1.26
		Neutral	1.02	1.00	1.05
		Positive	1.03	1.02	1.04
	Low	Negative	1.07	1.05	1.09
		Neutral	1.05	1.03	1.06
		Positive	0.94	0.92	0.95
RMSSD	High	Negative	−2.89	−2.90	−2.87
		Neutral	−2.94	−2.97	−2.92
		Positive	−3.02	−3.03	−3.00
	Low	Negative	−3.07	−3.09	−3.05
		Neutral	−3.10	−3.12	−3.09
		Positive	−3.06	−3.07	−3.04

**Table 7 sensors-20-03136-t007:** Descriptive statistics of vagal response depending on affective empathy level and arousal. M = Mean, SD = Standard deviation. 5% CI = 95% of confidence interval.

Vagal Response	Affective Empathy Level	Arousal Level	M	95% CI	
				Lower	Upper
RSA_PB	High	Arousal	6.53	6.49	6.56
		Neutral	6.40	6.35	6.44
		Relaxation	6.72	6.69	6.75
	Low	Arousal	6.60	6.57	6.63
		Neutral	6.87	6.85	6.90
		Relaxation	6.58	6.56	6.60
RSA_HF	High	Arousal	1.02	1.00	1.04
		Neutral	1.06	1.03	1.08
		Relaxation	1.05	1.03	1.07
	Low	Arousal	1.03	1.01	1.05
		Neutral	1.14	1.13	1.16
		Relaxation	0.99	0.97	1.00
RMSSD	High	Arousal	−3.05	−3.07	−3.03
		Neutral	−3.15	−3.17	−3.13
		Relaxation	−2.98	−3.00	−2.97
	Low	Arousal	−3.11	−3.13	−3.09
		Neutral	−2.92	−2.93	−2.90
		Relaxation	−3.04	−3.05	−3.03

**Table 8 sensors-20-03136-t008:** Descriptive statistics of vagal response depending on identification level and valence. M = Mean, SD = Standard deviation. 95% CI = 95% of confidence interval.

Vagal Response	Identification Level	Valence Level	M	95% CI	
				Lower	Upper
RSA_PB	High	Negative	6.65	6.62	6.69
		Neutral	6.53	6.49	6.56
		Positive	6.61	6.59	6.63
	Low	Negative	6.88	6.85	6.91
		Neutral	6.66	6.63	6.69
		Positive	6.57	6.54	6.59
RSA_HF	High	Negative	1.10	1.08	1.12
		Neutral	1.03	1.01	1.05
		Positive	1.04	1.03	1.05
	Low	Negative	1.19	1.17	1.20
		Neutral	1.05	1.03	1.07
		Positive	0.91	0.89	0.93
RMSSD	High	Negative	−3.03	−3.05	−3.01
		Neutral	−3.10	−3.12	−3.08
		Positive	−3.02	−3.04	−3.01
	Low	Negative	−2.95	−2.96	−2.93
		Neutral	−3.02	−3.04	−3.00
		Positive	−3.05	−3.07	−3.04

**Table 9 sensors-20-03136-t009:** Descriptive statistics of vagal response depending on identification level and arousal. M = Mean, SD = Standard deviation.

Vagal Response	Identification Empathy Level	Arousal Level	M	95% CI	
				Lower	Upper
RSA_PB	High	Arousal	6.53	6.49	6.56
		Neutral	6.33	6.28	6.38
		Relaxation	6.53	6.50	6.57
	Low	Arousal	6.76	6.73	6.80
		Neutral	6.69	6.66	6.71
		Relaxation	6.72	6.70	6.74
RSA_HF	High	Arousal	1.08	1.06	1.10
		Neutral	0.90	0.87	0.93
		Relaxation	0.97	0.95	0.99
	Low	Arousal	1.10	1.08	1.12
		Neutral	1.08	1.07	1.10
		Relaxation	1.03	1.01	1.04
RMSSD	High	Arousal	−3.08	−3.09	−3.06
		Neutral	−3.14	−3.16	−3.11
		Relaxation	−3.11	−3.12	−3.09
	Low	Arousal	−3.00	−3.02	−2.98
		Neutral	−2.98	−3.00	−2.97
		Relaxation	−2.99	−3.00	−2.98

**Table 10 sensors-20-03136-t010:** Descriptive statistics for vagal response depending on total empathy level and valence. M = Mean, SD = Standard deviation. 95% CI = 95% of confidence interval.

Vagal Response	Total Level	Valence Level	M	95% CI	
				Lower	Upper
RSA_PB	High	Negative	6.86	6.82	6.90
		Neutral	6.56	6.52	6.60
		Positive	6.60	6.57	6.62
	Low	Negative	6.74	6.71	6.77
		Neutral	6.63	6.59	6.66
		Positive	6.59	6.56	6.61
RSA_HF	High	Negative	1.21	1.19	1.24
		Neutral	1.01	0.98	1.03
		Positive	1.02	1.01	1.03
	Low	Negative	1.12	1.10	1.13
		Neutral	1.06	1.04	1.08
		Positive	0.95	0.93	0.96
RMSSD	High	Negative	−2.95	−2.97	−2.93
		Neutral	−3.05	−3.07	−3.03
		Positive	−3.02	−3.03	−3.01
	Low	Negative	−3.00	−3.01	−2.98
		Neutral	−3.06	−3.07	−3.04
		Positive	−3.05	−3.07	−3.04

**Table 11 sensors-20-03136-t011:** Descriptive statistics for vagal response depending on total empathy level and arousal. M = Mean, SD = Standard deviation. 95% CI = 95% of confidence interval.

Vagal Response	Total Empathy Level	Arousal Level	M	95% CI	
				Lower	Upper
RSA_PB	High	Arousal	6.47	6.44	6.50
		Neutral	6.49	6.44	6.54
		Relaxation	6.61	6.58	6.65
	Low	Arousal	6.68	6.65	6.71
		Neutral	6.76	6.73	6.79
		Relaxation	6.66	6.64	6.69
RSA_HF	High	Arousal	1.04	1.02	1.05
		Neutral	1.03	1.00	1.06
		Relaxation	0.99	0.97	1.01
	Low	Arousal	1.07	1.06	1.09
		Neutral	1.10	1.08	1.11
		Relaxation	1.02	1.01	1.03
RMSSD	High	Arousal	−3.10	−3.11	−3.08
		Neutral	−3.08	−3.11	−3.06
		Relaxation	−3.03	−3.05	−3.01
	Low	Arousal	−3.07	−3.08	−3.05
		Neutral	−2.95	−2.97	−2.94
		Relaxation	−3.01	−3.02	−3.00

**Table 12 sensors-20-03136-t012:** Descriptive statistics: Self-reporting empathy scores depending on existence of co-viewer and valence level. M = Mean, SD = Standard deviation. 95% CI = 95% of confidence interval.

Empathy Scores	Co-Viewer	Valence Leve	M	95%CI	
				Lower	Upper
Cognitive empathy	Yes	Negative	4.01	3.79	4.24
		Neutral	4.16	3.91	4.41
		Positive	4.42	4.24	4.60
	No	Negative	4.43	4.19	4.68
		Neutral	4.21	3.97	4.45
		Positive	4.52	4.34	4.69
Affective empathy	Yes	Negative	2.26	1.89	2.64
		Neutral	2.04	1.62	2.45
		Positive	2.65	2.36	2.95
	No	Negative	3.08	2.68	3.48
		Neutral	2.54	2.14	2.93
		Positive	3.44	3.16	3.73
Identification	Yes	Negative	2.68	2.31	3.04
		Neutral	3.21	2.81	3.62
		Positive	3.24	2.95	3.53
	No	Negative	3.21	2.82	3.60
		Neutral	3.33	2.95	3.71
		Positive	3.93	3.65	4.21
Total empathy	Yes	Negative	3.05	2.80	3.29
		Neutral	3.23	2.96	3.50
		Positive	3.50	3.31	3.70
	No	Negative	3.61	3.35	3.87
		Neutral	3.43	3.17	3.69
		Positive	4.01	3.82	4.19

**Table 13 sensors-20-03136-t013:** Descriptive statistics: Self-reporting empathy scores depending on existence of co-viewer and arousal level. M = Mean, SD = Standard deviation.

Empathy Scores	Co-Viewer	Valence Level	M	95% CI	
				Lower	Upper
Cognitive empathy	Yes	Arousal	4.52	4.28	4.76
		Neutral	4.74	4.42	5.06
		Relaxation	3.94	3.70	4.18
	No	Arousal	4.28	4.06	4.50
		Neutral	4.25	4.07	4.43
		Relaxation	4.41	4.25	4.57
Affective empathy	Yes	Arousal	2.80	2.40	3.20
		Neutral	3.94	3.41	4.48
		Relaxation	2.19	1.79	2.59
	No	Arousal	2.95	2.59	3.32
		Neutral	2.28	1.98	2.57
		Relaxation	3.00	2.73	3.26
Identification	Yes	Arousal	3.45	3.05	3.85
		Neutral	4.07	3.55	4.60
		Relaxation	2.98	2.58	3.37
	No	Arousal	3.40	3.04	3.77
		Neutral	2.92	2.62	3.21
		Relaxation	3.57	3.31	3.84
Total empathy	Yes	Arousal	3.66	3.39	3.92
		Neutral	4.27	3.93	4.62
		Relaxation	3.11	2.85	3.37
	No	Arousal	3.60	3.36	3.84
		Neutral	3.22	3.03	3.41
		Relaxation	3.71	3.54	3.89

**Table 14 sensors-20-03136-t014:** Descriptive statistics for vagal response depending on existence of co-viewer and valence.

Vagal Response	Co-Viewer Condition	Valence Level	M	95%CI	
				Lower	Upper
RSA_PB	With	Negative	6.73	6.70	6.76
		Neutral	6.62	6.58	6.65
		Positive	6.61	6.58	6.63
	Without	Negative	6.84	6.80	6.87
		Neutral	6.58	6.55	6.62
		Positive	6.58	6.56	6.60
RSA_HF	With	Negative	1.16	1.14	1.18
		Neutral	1.08	1.06	1.10
		Positive	1.03	1.01	1.04
	Without	Negative	1.13	1.11	1.15
		Neutral	1.01	0.99	1.02
		Positive	0.96	0.94	0.97
RMSSD	With	Negative	−3.00	−3.01	−2.98
		Neutral	−3.07	−3.09	−3.05
		Positive	−3.03	−3.04	−3.01
	Without	Negative	−2.97	−2.99	−2.95
		Neutral	−3.04	−3.06	−3.02
		Positive	−3.04	−3.06	−3.03

**Table 15 sensors-20-03136-t015:** Descriptive statistics for vagal response depending on existence of co-viewer and arousal.

Vagal Response	Co-Viewer Condition	Arousal Level	M	95% CI	
				Lower	Upper
RSA_PB	With	Arousal	6.42	6.39	6.46
		Neutral	6.57	6.53	6.62
		Relaxation	6.72	6.68	6.75
	Without	Arousal	6.61	6.57	6.64
		Neutral	6.73	6.70	6.75
		Relaxation	6.69	6.66	6.71
RSA_HF	With	Arousal	1.01	0.99	1.03
		Neutral	1.07	1.05	1.10
		Relaxation	1.07	1.05	1.09
	Without	Arousal	1.01	1.00	1.03
		Neutral	1.12	1.10	1.13
		Relaxation	1.00	0.98	1.01
RMSSD	With	Arousal	−3.13	−3.15	−3.11
		Neutral	−3.02	−3.05	−3.00
		Relaxation	−3.07	−3.09	−3.06
	Without	Arousal	−3.03	−3.05	−3.02
		Neutral	−2.95	−2.96	−2.94
		Relaxation	−3.02	−3.03	−3.00

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
