# Peer review of "Vagal Tone Differences in Empathy Level Elicited by Different Emotions and a Co-Viewer"

_sensors, 2020, doi:10.3390/s20113136_

Round 1

Reviewer 1 Report

The paper proposes an interesting approach for determining the degree of empathy elicited by different emotions. However, the way in which the article is written is full of flaws.

First of all, the title is too long (19 words), contains a large number of words/concepts/ideas (empathy, emotions, co-viewer, vagal tone, multimedia) and is very difficult to understand the core purpose of the paper. I highly recommend to shorten it and to find a way to connect the concepts covered in the paper in a more compact form, in maximum 10-12 words. The title should be succinct, free of obscure abbreviations and to the point, and it should describe key content in an effective way. The abstract should be clear, interesting, understandable, accurate, specific and to the point. Ensure that your title and abstract do not misrepresent your research or mislead the reader.

The abstract, as well as the rest of the paper is full of grammar errors. The phrases are too long, the ideas are linked inappropriately, there are many flaws in the content description. I will mention only a few, but there are many other more. My strong recommendation is to find a native English speaker to help you with the proofreading because in the present form, this paper is UNPUBLISHABLE.

 “Empathy with multimedia content has been gathered research interest” –> “Empathy with multimedia content has gathered research interest”

The first phrase from the Abstract is too long and incomprehensible.

“Therefore, this study was to determine empathy level elicited by different emotions and existence of co-viewers using vagal tone” -> “existence of co-viewers” ?? I do not understand what you mean

“makes easier to catch empathetic responses” -> the appropriate verb here is “collect”. So, “collect empathetic responses”

“4 videos which eliciting different level of valence and arousal” -> “4 videos which elicited different levels of valence and arousal”

“they answered both valence and arousal level to figure out which emotion they empathized with” -> “they reported both valence and arousal levels…”. “to figure out which emotion they empathized with” -> I do not understand what you mean

“vagal tone was significantly different according to empathy level but significant patterns were different according to each empathy factors and emotions” -> totally incomprehensible

“higher vagal response in almost all emotions that participants empathized with (p < 0.05)” -> really bad use of English

“The results suggested emotion and cognitive factors that people empathized with should be considered to evaluate empathy effectively” -> really bad use of English

So, the examples can continue all throughout the paper. It seems as the authors translated the whole paper with Google Translate.

Regarding the scientific content:

The Introduction chapter is a mix of information. Instead of presenting the context of the research and the purpose of the experiment, you wrote a state of the art. The state of the art should be presented in a separate chapter. You did not event define the concepts of “empathy”, “emotion” and “vagal tone”. They are very important in order to familiarize the reader with the topic, assuming that not all the readers are experts in the field. In the Introduction chapter, you should describe the structure of the paper: chapter II presents, chapter III presents…

I would like to find out more about the preprocessing of the data. It is a very important step and this section should be treated more seriously.

The size of the font used in Tables 3-12 is too big. Please respect the template formatting.

There are no references in the text to the tables. You should describe in the text the content presented in each table.

There are no references in the text to the figures. Describe each figure and each plot properly in order to help the reader understand your results.

The results are not discussed thoroughly. There are no references to similar experiments and research pursuits. You did not present the novelty of your results by comparison with other experiments. I suggest to add a table to present the pros and cons of your research, limitations and constraints as well as those of others.

You should strengthen the conclusions section. You should add a reference to future research ideas - what you plan to do next, why are useful your results, in which field they can be used further on.

Author Response

Response to Reviewer 1 Comment

Point 1: First of all, the title is too long (19 words), contains a large number of words/concepts/ideas (empathy, emotions, co-viewer, vagal tone, multimedia) and is very difficult to understand the core purpose of the paper. I highly recommend to shorten it and to find a way to connect the concepts covered in the paper in a more compact form, in maximum 10-12 words. The title should be succinct, free of obscure abbreviations and to the point, and it should describe key content in an effective way. The abstract should be clear, interesting, understandable, accurate, specific and to the point. Ensure that your title and abstract do not misrepresent your research or mislead the reader.

Response 1: Thank you for your kind feedback. I shortened the title in 12 words by excluding the word multimedia which can confuse the readers.

Point 2: The abstract, as well as the rest of the paper is full of grammar errors. The phrases are too long, the ideas are linked inappropriately, there are many flaws in the content description. I will mention only a few, but there are many other more. My strong recommendation is to find a native English speaker to help you with the proofreading because in the present form, this paper is UNPUBLISHABLE.

Response 2: I rewrote abstract and corrected grammar errors in the whole paper as much as I can with help of native speakers.

Point 3: The Introduction chapter is a mix of information. Instead of presenting the context of the research and the purpose of the experiment, you wrote a state of the art. The state of the art should be presented in a separate chapter. You did not event define the concepts of “empathy”, “emotion” and “vagal tone”. They are very important in order to familiarize the reader with the topic, assuming that not all the readers are experts in the field. In the Introduction chapter, you should describe the structure of the paper: chapter II presents, chapter III presents…

Response 3: I divided introduction per information as you pointed out. In the final section of introduction, I summarized the limitation of the state of the art and the objective of the present study. (119-127 line)

Point 4: I would like to find out more about pre-processing of the data. It is a very important step and this section should be treated more seriously.

Response 4: I added more information about preprocessing such as normal range of RRI and how to handle missing RRI value. (191-195 line)

Point 5: the size of the font used in Tables 3-12 is too big. Please respect the template formatting.

Response 5: I doubled checked but the font size was same with template formatting (Font size :10). If it is wrong, please let me know .

Point 6: There are no references in the text to the tables. You should describe in the text the content presented in each table.

Response 6: I added the reference in the text to all tables and describe the tables’ content

Point 7: There are no references in the text to the figures. Describe each figure and each plot properly in order to help the reader understand your results

Response 7: I added the reference in the text to all tables and describe the figures’ content

Point 8: The results are not discussed thoroughly. There are no references to similar experiments and research pursuits. You did not present the novelty of your results by comparison with other experiments. I suggest to add a table to present the pros and cons of your research, limitations and constraints as well as those of others.

Response 8: I rewrote discussion and added similar studies with mine and compared them with the present study. Pros of my research included in discussion and conclusion. Cons of my research was presented in limitation section

Point 9: You should strengthen the conclusions section. You should add a reference to future research ideas - what you plan to do next, why are useful your results, in which field they can be used further on.

Response 9: I rewrote conclusion and added future reference idea

Reviewer 2 Report

Dear Authors,

With interest I was reading your paper, The text is well written, the univariate statistics is difficult to read. We have the following critical comments.

  1. The statistical analysis is based on elementary univariate statistics. The authors have to use more advanced multivariate statistics to make the paper more interesting. First we are interested to see what the correlation between different items of the test, the authors have to apply a homogeneity analysis to see if groups of respondents give similar answers to the questions, a factor analysis can be applied to see the relation between 3 factors of empathy.
  2. We like to see plots of repondents for example in 2 dimensions with cognitive and affective scores
  3. The experimental procedure is not clear, in figure 2 the experimental setting has been explained. From figure 2 we can read that the experiments start with an assessment of an emotion quotiënt.
  4. The authors select 4 videos out of 5 in the pre-test. The selection of 4 is not clear also it is not clear why they represent the four domains. We like to see the arousal/valence levels scores of the 198 respondents.
  5. The selection of the 4-5 videos is not clear with respect to supposed applications. It is also not clear why there are positive and negative scores with the stimuli.
  6. The authors chose videos with extreme emotions. It is not clear what will happen in case daily life emotions are used
  7. ECG was applied, but we expect a lot missing values because of failing technology, nothing is reported.
  8. The authors stress the use of ECG. In many cases emotions are assessed by facial expressions. maybe this less reliable but non invasive

Author Response

Point 1: The statistical analysis is based on elementary univariate statistics. The authors have to use more advanced multivariate statistics to make the paper more interesting First we are interested to see what the correlation between different items of the test, the authors have to apply a homogeneity analysis to see if groups of respondents give similar answers to the questions, a factor analysis can be applied to see the relation between 3 factors of empathy.

Response 1: Thank you for your kind feedback. But I think I made you confused. Correlation analysis and factor analysis between different items of the test was already analysed in the Korean paper (reference no. 13) that developed the empathy questionnaire this study used.  According to  [13], each empathy factors were independent. Exploratory and confirmation factor analysis already done in [13]. I thought the reason you recommended homogeneity analysis was for the validation of empathy questionnaire. If it is right I think it also done in [13].

Point 2: We like to see plots of repondents for example in 2 dimensions with cognitive and affective scores

Response 2: I’m really sorry but I did not understand what you mean. If you give me more explanation, I will reflect your feedback.

Point 3: The experimental procedure is not clear, in figure 2 the experimental setting has been explained. From figure 2 we can read that the experiments start with an assessment of an emotion quotiënt.

Response 3: I’m sorry. I made mistake when I make Figure 1(flow chart). I updated the figure 1.  Paper in the Figure 2(experiment setting) is for the self-reporting for empathy valence, arousal and scores. Participants were asked to answer the questionnaire after they watched per stimulus. Researchers watched them whether they follow our instruction.

Point 4: 4. The authors select 4 videos out of 5 in the pre-test. The selection of 4 is not clear also it is not clear why they represent the four domains. We like to see the arousal/valence levels scores of the 198 respondents.

Response 4: I added the process of how we choose 4 videos out of 5 (141~144 line). Also, mean valence and arousal scores of 198 respondents were added in Table 1. (I updated Table 1)

Point 5: The selection of the 4-5 videos is not clear with respect to supposed applications. It is also not clear why there are positive and negative scores with the stimuli.

Response 5: The 4 videos were presented to elicit various emotions (More precisely, quadrants of circumplex model of affect). I added the explanation of quadrants of circumplex model in 60~67 line.

Since this paper was to investigate vagal differences according to empathy levels elicited by different emotions, we intended participants felt empathy with various emotions

Point 6: The authors chose videos with extreme emotions. It is not clear what will happen in case daily life emotions are used

Response 6: Although videos were presented to elicit quandrants of circumplex model, it is difficult to assure that participants really feel emotion that we intended. Therefore, this study categorized participants’ assessment into three group per each emotional dimension to reflect their genuine emotions. So this study included daily emotion and extreme emotion. (150~ 153 lines)

Point 7: ECG was applied, but we expect a lot missing values because of failing technology, nothing is reported.

Response 7: I added following information in the preprocessing session: how to handle missing RRI and such as normal range of RRI. (188-192)

Point 8: 8. The authors stress the use of ECG. In many cases emotions are assessed by facial expressions. maybe this less reliable but non invasive

Response 8: Facial expression also have been measured to detect empathy as you said because one of the empathetic responses is facial mimicry but facial expression is more appropriate with interpersonal empathy. I wrote why this study did not use facial expression in the paper (86~90 lines).

Round 2

Reviewer 1 Report

I appreciate the fact that you tried to improve the quality of your paper. I still found English errors all throughout the paper. I recommend extensive editing of English language and style. 

Author Response

Thank you for your feedback

I revised grammar errors in the whole manuscript with help of native speaker

I also changed statistical analysis (multiple t-test -> 2-way manova)

Consequently I changed section 2.7 statistical analysis, Results, Discussions and Conclusion. 

Author Response

Thank your for your detailed feedback

Point 1: 1.Pre test

The authors report about a pre-test of 198 respondents. The valence and arousal components are computed but it is not explained how this is done

Response 1: I added the way to collect valence and arousal scores in 143~145 lines.

Point 2: 62 respondents in the experiment have to report their emotions but again it is not clear how this is done

Response 2: I added the way to collect valence and arousal scores in 157~159 lines.

(By the way, participant of the main experiment is 59. We recruited 62 but we were not able to use data of 3 participants due to technical failure. I'm sorry to make you confused. This information was described section 2.1.)

Point 3: 3. To compute empathy, the authors use a questionnaire developed by a Korean research [13]. We were not able to verify this reference because we don’t master the Korean. We have our doubt if it is possible to compute empathy using 13 items. The term vicarious is also not clear and how this can be mapped on cognitive and affective empathy. We have out doubts about the tree concepts and the foundation in the scientific literature. The authors of the current paper have to verify the concepts

Response 3: The point of cognitive empathy is to understand or perceive other's situation. Therefore the ‘empathic understanding’ is expected to explain cognitive empathy (The author also explained empathic understanding is corresponding to cognitive empathy).  I changed the word into cognitive empathy since cognitive empathy is more frequently used. Also the [13] referenced the following paper explaining cognitive empathy

  1. (R. Horgan, “Development of an empathy scale,” J. of Consulting and Clinical Psychology, Vol.33, 307-316, 1969.)

In case of affective empathy, the key point is individuals can feel other's emotion as if it is your emotions even though that emotion is not yours. In my opinion, reason why [13] used the word 'vicarious' was people feel emotion which is not their own when they felt affective empathy. I also added information that [13]’s author’s reference paper to explain vicarious response (affective, emotional empathy)

  1. Mehrabian and N. Epstein, “A measure of emotional empathy,” J. of Personality, Vol.40, pp.525-543, 1972.
  2. Smith, “Cognitive empathy and emotional empathy in human behavior and evolution,”Psychological Record, Vol.56, No.1, pp.3-21, 2006.

The concept of empathy was explained in section 1.1

Point 4: The authors state in their paper they computed the CERA score. It is not too complicated to compete the correlation between the 11 items. Then we can split up the correlation matrix. The three 3x3 blocks among the main diagonal correspond to cognitive, affective and identification empathy and should have high valued correlation coefficients, much higher that the correlation outside the diagonal.

Response 4: I conducted factor analysis and correlation test to verify CERA scales using data of 198 pre-test participants. Figure 1 presented correlation matrix of the CERA scale

Point 5: In line 163, the authors state 'the selected 11 items were…???'

Response 5: [13] collected 17 items initially and chose the final 11items to construct the CERA scale. That's why I mentioned the selected 11items. I'm sorry to make you confused. I added the general explanation of the CERA scale in section 2.4.1)

Point 6: The internal consistency can also be computed using Croanbach’s alpha coefficient and computing, showing that there are indeed three underlying factors in the questionnaire of Korean researcher. We have serious doubts and this should be verified by available data

Response 6: I conducted factor analysis and correlation test to verify CERA scales using data of 198 pre-test participants. I added the factor analysis result in section 2.4.1 and Table 3.  Figure 1 presented correlation matrix of the CERA scale

Point 7: As said earlier, we don't prefer the univariate statistics as displayed in section 3. We proposed already to plot the cognitive and affective empathy coefficient of every respondent and verify if the cloud of respondents has a homogeneous distribution or is split up in clusters or whatever.

Response 7: I conducted homogenous test for 2-way MANOVA to investigate the relation between empathy, emotion and vagal response. (empathy levels (high vs low) x emotion categories (negative vs neutral vs positive) 8 times. (Since emapthy level was measured though cognitive, affective and identification and emotion were measured in valence and arousal dimension). In case that data violated the assumption of homogeneity, I ran Pillai's test that does not assume homogeneity and scheffe’s post hoc test for individual ANOVAs. I revised the section 2.7 and Result section in general.

Point 8: In 3.1.1 the authors state that detailed statistics are presented in Table 3. The authors leave it to the readers to give an interpretation of the data but this should be done by the authors

Response 8: I changed Result section

Point 9: It is still not clear how the authors handle missing data and overlapping data. The data in table 6,7 are to clean

Response 9: Missing RRI were substituted mean of former normal RRI values so there were no missing RSA_PB, RSA_HF and RMSSD (248-249 line). But still there were outliers of RSA_HF, RMSSD (there were no outliers in RSA_PB) so I dropped them. (251 line)

Response 10: The section conclusion is very short and not based on facts. In the conclusion we like to read if the aims of the research or realised or not

Response 10: I rewrote the conclusion section.
